# Antagonistic interactions can stabilise fixed points in heterogeneous linear dynamical systems

**Samuel Cure[1⋆] and Izaak Neri[2]**

**1** Okinawa Institute of Science and Technology, 1919-1 Tancha,
Onna-son, Okinawa 904-0495, Japan
**2** Department of Mathematics, King's College London,
Strand, London, WC2R 2LS, UK

⋆ samuel.cure@oist.jp

## Abstract

We analyse the stability of large, linear dynamical systems of variables that interact through a fully connected random matrix and have inhomogeneous growth rates. We show that in the absence of correlations between the coupling strengths, a system with interactions is always less stable than a system without interactions. Contrarily to the uncorrelated case, interactions that are antagonistic, i.e., characterised by negative correlations, can stabilise linear dynamical systems. In particular, when the strength of the interactions is not too strong, systems with antagonistic interactions are more stable than systems without interactions. These results are obtained with an exact theory for the spectral properties of fully connected random matrices with diagonal disorder.



# 1   Introduction

We consider a dynamical system described by $n$ variables $x_j \in \mathbb{R}$ that are labeled by indices $j = \{1, 2, \ldots, n\} = [n]$ and where $t \in \mathbb{R}^+$ is the time index. The evolution in time of the variables $x_j(t)$ is described by a set of randomly coupled, linear differential equations of the form

$$\partial_t x_j(t) = \sum_{k=1}^{n} A_{jk} x_k(t), \tag{1}$$

where the $A_{jk}$ are the entries of a random matrix $\mathbf{A}$ of dimension $n \times n$. The fixed point $\vec{x} = 0$ of the set of Eqs. (1) is stable when all eigenvalues of $\mathbf{A}$ have negative real parts. On the other hand, if there exists at least one eigenvalue with a positive real part, then the fixed point is unstable.

Differential equations of the form Eq. (1) appear in linear stability analyses of complex systems described by nonlinear differential equations of the form $\partial_t \vec{y}(t) = \vec{f}(\vec{y}(t))$ where $\vec{y} = (y_1, y_2, \ldots, y_n)$. For example, in theoretical ecology ecosystems are modelled with Lotka-Volterra equations, where the variable $\vec{y}$ quantify the population abundances of the different species in the population [1]. Other examples are models for neural networks, for which $\vec{y}$ represents the neuronal firing rates or the membrane potentials [2–4], and models of economies [5], for which $\vec{y}$ represents economic variables such as the prices of goods. If the differential system determined by $\vec{f}$ admits a fixed point, defined as $\vec{f}(\vec{y}^*) = 0$, then the dynamics of $\vec{x} = \vec{y} - \vec{y}^*$ near the fixed point is given by Eq. (1), where $\mathbf{A}$ is the Jacobian of $\vec{f}$. The linear stability of a complex system that settles in a fixed point state is thus determined by the real part of the *leading eigenvalue* $\lambda_1$, which is defined as an eigenvalue of the Jacobian matrix $\mathbf{A}$ that has the largest real part.

To study complex systems, Wigner [6], Dyson [7] and May [8], among others, suggested to study random matrices $\mathbf{A}$, and the task at hand is then to determine the real part of the leading eigenvalue as a function of the parameters that define the random matrix ensemble. Although one should be careful in drawing conclusions about the dynamics of nonlinear systems from

the study of randomly coupled linear differential equations, random matrix theory has the advantage of providing analytical insights about the influence of interactions on linear stability. In fact, linear stability analyses with random matrix theory have been used to study the onset of chaos in random neural networks [2–4], the stability of ecosystems modelled by Lotka-Volterra equations with random interactions [8–14], economies [15], or gene regulatory networks [16]. Moreover, although traditionally random matrix models are fully connected, recently exact results have been derived for the stability of linear models defined on complex networks [17–19].

So far, stability analyses for randomly coupled, linear dynamical systems have mainly focused on matrices **A** with diagonal entries, which we also call the growth rates, that are fixed to a constant value $d$, i.e., [8]

$$A_{jk} = \frac{J_{jk}}{\sqrt{n}}(1 - \delta_{j,k}) + d\delta_{j,k}, \tag{2}$$

where $\delta_{j,k}$ is the Kronecker delta function, and where the coupling strengths $J_{jk}$ are random variables drawn from a certain distribution. Following Refs. [9,20], we consider the case where the pairs of random variables $(J_{jk}, J_{kj})$ are independent and identically (i.i.d.) distributed random variables drawn from a distribution with

$$\langle J_{ij} \rangle = 0, \quad \langle J_{ij}^2 \rangle = \sigma^2, \quad \text{and} \quad \langle J_{ij} J_{ji} \rangle = \tau\sigma^2, \tag{3}$$

where the variance $\sigma^2$ of the entries $J_{ij}$ quantifies the strength of the interactions, and $\tau \in [-1, 1]$ is the Pearson correlation coefficient between the variables $J_{jk}$ and $J_{kj}$. The sign of the parameter $\tau$ is important in theoretical ecology as it determines the nature of the trophic interactions between two species. If the interactions are on average competitive ($J_{ij} < 0$ and $J_{ji} < 0$) or mutualistic ($J_{ij} > 0$ and $J_{ji} > 0$), then $\tau > 0$. On the other hand, if the interactions are on average antagonistic ($J_{ij} > 0$ and $J_{ji} < 0$ or $J_{ij} < 0$ and $J_{ji} > 0$), then $\tau < 0$ [9–11,19]. In theoretical ecology, antagonistic interactions are also called predator-prey interactions as they describe trophic interactions between two species for which one predates on the other.

The leading eigenvalue of random matrices of the form (2) is given by [8,21–23]

$$\mathrm{Re}(\lambda_1) = \sigma(1 + \tau) + d. \tag{4}$$

It follows from Eq. (4) that in the case of homogeneous relaxation rates $d < 0$ is required for a linear system to be stable. Hence, when the diagonal entries of **A** are fixed to a constant value $d$, then interactions $J_{jk}$ always destabilise fixed points in large dynamical system.

In the model given by Eq. (2) it holds that in the absence of interactions ($J_{ij} = 0$) either all variables are stable (when $d < 0$) or all variables are unstable (when $d > 0$). In this paper, we relax this condition and consider random matrix models with growth rates $A_{jj} = D_j$ that fluctuate from one variable to the other. In the symmetric case ($\tau = 1$), such random matrices are called deformed Wigner matrices [24–26] and in this case a functional equation that determines the spectral distribution in the limit of large $n$ has been derived by Pastur in Ref. [24]. Another case that has been studied in the literature is when **A** is the adjacency matrix of a random directed graph with diagonal disorder [17,18,27], which corresponds in the dense limit with $\tau = 0$ [28], and for which a simple equation for the boundary of the spectrum as a function of the distribution of diagonal matrix entries has been derived.

On the other hand, in the present paper we focus on the case of heterogeneous relaxation rates $D_j$ with negative $\tau$, which is, as discussed, of particular interest for ecology. This case has been studied significantly less in the literature, a notable exception being Ref. [29]. Here, apart from completing the theory of Ref. [29] by deriving analytical results for eigenvalue outliers, which are important when considering the leading eigenvalue, we also show that for

negative $\tau$ the leading eigenvalue can be negative, even if a finite fraction of the relaxation rates $D_j$ are positive. The latter finding, not discussed in Ref. [29], implies that antagonistic interactions can stabilise linear systems when the interactions are neither too weak nor too strong, even if a finite fraction of the variables are unstable in isolation, and this constitutes the main result of this paper.

The paper is organised as follows. In Sec. 2 we define the model that we study, which is a fully connected random matrix with diagonal disorder. In Sec. 3, we discuss the cavity method, which is a method from theoretical physics that we use to study the model in the limit of infinitely large random matrices. In Sec. 4, we present the main results for the boundary for the spectrum of fully connected matrices with diagonal disorder and in Sec. 5 we present analytical results for the eigenvalue outlier. In Sec. 6 we use the obtained theoretical results to derive phase diagrams for the linear stability of fixed points. We end the paper with a discussion in Sec. 7. The paper also contains a few appendices where we present details about the mathematical derivations.

## 2 Fully connected random matrices with diagonal disorder

We consider the random matrix model

$$A_{jk} = \frac{J_{jk}}{\sqrt{n}}(1 - \delta_{j,k}) + \frac{\mu}{n} + D_j \delta_{j,k}, \tag{5}$$

where the (off-diagonal) pairs $(J_{ij}, J_{ji})$ are i.i.d. random variables drawn from a joint distribution $p_{J_1,J_2}$ with moments as specified in the Eqs. (3), where the diagonal elements $D_j$ are i.i.d. random variables drawn from a distribution $p_D$, and where $\mu \in \mathbb{R}$ is a constant shift of the matrix elements. Note that without loss of generality we have set $\langle J_{ij} \rangle = 0$, as a nonzero average value can be incorporated into the parameter $\mu$.

As will become clear later, just as is the case for the circular law [30, 31], in the limit of $n \gg 1$ the boundary of the spectrum of **A** is a deterministic curve in the complex plane that depends on the distribution $p_{J_1,J_2}$ of $(J_{ij}, J_{ji})$ only through its first two moments given in Eq. (3), and hence we will not need to specify $p_{J_1,J_2}$. On the other hand, the boundary of the spectrum of **A** depends in a nontrivial way on the distribution $p_D$, and therefore it will be interesting to study the effect that the shape of $p_D$ has on the leading eigenvalue. Due to the constant shift $\mu$, the spectrum may also contain a single (deterministic) eigenvalue outlier in the limit of large $n \gg 1$ [17, 27, 32, 33].

In the special case when $p_D(x) = \delta(x - d)$ we recover the model given by Eq. (4). A more interesting case is when the growth rates $D_j$ are heterogeneous, and arguably the most simple model for heterogeneous growth rates considers that the $D_j$ can take two possible values, yielding a bimodal distribution

$$p_D(x) = p\,\delta(x - d_-) + (1 - p)\,\delta(x - d_+), \tag{6}$$

with $d_- < 0$, $d_+ > 0$, and $p \in [0, 1]$, and where $\delta(x - d)$ denotes the Dirac delta distribution. In this example, a fraction $(1 - p)$ of variables $x_j$ are unstable in the absence of interactions ($\sigma^2 = 0$). We also consider cases where $p_D$ is a continuous distribution. One example of a continuous distribution is the uniform distribution defined on an interval $[d_-, d_+]$, i.e.,

$$p_D(x) = \begin{cases} 0, & \text{if } x \notin [d_-, d_+], \\ \frac{1}{d_+ - d_-}, & \text{if } x \in [d_-, d_+]. \end{cases} \tag{7}$$

Since the uniform distribution is supported on a bounded set, we will also consider an example

for which $p_D$ has unbounded support, namely, we will consider the Gaussian distribution

$$p_D(x) = \frac{1}{\sqrt{2\pi}} e^{-\frac{x^2}{2}} \,, \tag{8}$$

with zero mean and unit variance.

The main question we address in this paper is whether the interaction variables $J_{ij}$ can stabilise a linear dynamical system even when a finite fraction of variables are unstable in the absence of interactions, i.e., a finite fraction of species $i \in [n]$ have a positive growth rate $D_i$. In other words, we ask whether it is possible to have $\mathrm{Re}\,\lambda_1 < 0$ even when there exists a value $d > 0$ such that $p_D(d) > 0$.

## 3  Cavity method for the empirical spectral distribution of infinitely large matrices

We determine the leading eigenvalue $\lambda_1$ in the case of $\mu = 0$, when the spectrum of $\mathbf{A}$ has no outliers in the limit of $n \to \infty$.

In this case, the leading eigenvalue of the adjacency matrices $\mathbf{A}$ is determined by the empirical spectral distribution $\rho$ of the eigenvalues $\lambda_j$ of $\mathbf{A}$, defined by

$$\rho(z) = \lim_{n \to \infty} \frac{1}{n} \left\langle \sum_{j=1}^{n} \delta\left(x - \mathrm{Re}(\lambda_j)\right) \delta\left(y - \mathrm{Im}(\lambda_j)\right) \right\rangle \,, \tag{9}$$

for all $z = x + \mathrm{i}y \in \mathbb{C}$. The spectral distribution determines the leading eigenvalue of the continuous part of the spectrum through

$$\lambda_1 = \mathrm{argmax}_{\{z \in \mathbb{C}: \rho(z) > 0\}} \mathrm{Re}(z) \,. \tag{10}$$

Equation (10) holds as long as the spectrum of $\mathbf{A}$ does not have eigenvalue outliers [17,27], which for the model defined in Sec. 2 is the case as long as $\mu = 0$ [17,27].

The convergence in Eq. (9) should be understood as weak convergence [31], which implies that the average of any bounded and continuous function $f(z)$ defined on the complex plane converges in the limit of large $n$ to $\int_{\mathbb{C}} \mathrm{d}z\, \rho(z) f(z)$. Also, we can drop the average in the right-hand side of Eq. (9) as the spectral distribution converges almost surely and weakly to $\rho$ [31], and hence also the leading eigenvalue $\lambda_1$ as defined in Eq. (10) is a deterministic variable for large values of $n$.

The limiting distribution $\rho$ of random matrix models as defined in Sec. 2 have been studied before in several special cases. Notably, for the symmetric case with $\tau = 1$ Pastur derived a functional equation that determines $\rho$ [24]. Recently, the symmetric case was revisited in [26], and in that reference also the large deviations of $\lambda_1$ were computed in the case when the matrix entries $J_{ij}$ are drawn from a Gaussian distribution; note that large deviations are not universal and depend on the statistics of $(J_{ij}, J_{ji})$ as determined by the distribution $p_{J_1, J_2}$. In the case when $\tau = 0$ and $p_D$ is a bimodal distribution the spectral distribution $\rho$ has been determined in Refs. [34,35] and the $\tau = 0$ case for general $p_D$ has been considered in [28]. For random directed graphs with a prescribed distribution of indegrees and outdegrees, which corresponds with the case $\tau = 0$ in the limit of large mean degrees, a simple equation was derived for the boundary of the spectrum in Refs. [17,18,27]. Lastly, Ref. [29] obtained analytical results for the spectrum when $\tau < 0$ and $\mu = 0$.

We determine the spectral density $\rho(z)$ from the resolvent of the matrix $\mathbf{A}$, which can be determined with the cavity method [36,37]. The resolvent is defined as

$$\mathbf{G}(z) = (z\mathbf{1}_n - \mathbf{A})^{-1} \,, \quad z \notin \{\lambda_1, \lambda_2, \ldots, \lambda_n\} \,, \tag{11}$$

where $\mathbf{1}_n$ is the identity matrix of size $n$. The spectral distribution can be expressed in terms of the diagonal elements of the resolvent by [34]

$$\rho(z) = \lim_{n\to\infty} \frac{1}{\pi n} \partial^* \mathrm{Tr}\mathbf{G}(z), \quad \text{where} \quad \partial^* = \frac{1}{2}\frac{\partial}{\partial x} + \frac{\mathrm{i}}{2}\frac{\partial}{\partial y}. \tag{12}$$

For non-Hermitian matrices, the eigenvalues are in general complex-valued, and therefore in the limit of $n \to \infty$ we cannot get $\rho(z)$ from $\mathrm{Tr}\mathbf{G}(z)$ [23]. To overcome this, we use the Hermitization method [34] that considers the enlarged $2n \times 2n$ matrix

$$\mathbf{H} = \begin{pmatrix} \eta\mathbf{1}_n & z\mathbf{1}_n - \mathbf{A} \\ z^*\mathbf{1}_n - \mathbf{A}^T & \eta\mathbf{1}_n \end{pmatrix}, \tag{13}$$

where we have introduced a regulator $\eta$ that keeps all quantities well-defined in the limit of large $n$, where $\mathbf{A}^T$ is the transpose of the matrix $\mathbf{A}$, and where $z^*$ is the complex conjugate of $z$. The inverse of the matrix $\mathbf{H}$ is

$$\mathbf{H}^{-1} = \begin{pmatrix} \frac{\eta}{\eta^2\mathbf{1}_n - \mathbf{I}_l} & -(\eta^2\mathbf{1}_n - \mathbf{I}_l)^{-1}(z\mathbf{1}_n - \mathbf{A}) \\ -(z^*\mathbf{1}_n - \mathbf{A}^T)(\eta^2\mathbf{1}_n - \mathbf{I}_l)^{-1} & \frac{\eta}{\eta^2\mathbf{1}_n - \mathbf{I}_r} \end{pmatrix}, \tag{14}$$

where

$$\mathbf{I}_l = (z\mathbf{1}_n - \mathbf{A})(z^*\mathbf{1}_n - \mathbf{A}^T) \quad \text{and} \quad \mathbf{I}_r = (z^*\mathbf{1}_n - \mathbf{A}^T)(z\mathbf{1}_n - \mathbf{A}). \tag{15}$$

In the limit $\eta \to 0$, we obtain

$$\mathbf{H}^{-1} = \begin{pmatrix} \mathbf{0}_n & \mathbf{G}(z) \\ \mathbf{G}(z) & \mathbf{0}_n \end{pmatrix} - \eta \begin{pmatrix} \mathbf{I}_l^{-1} & -\mathbf{0}_n \\ \mathbf{0}_n & \mathbf{I}_r^{-1} \end{pmatrix} + \eta^2 \begin{pmatrix} \mathbf{0}_n & \mathbf{G}(z)\mathbf{I}_l^{-1} \\ \mathbf{G}(z)\mathbf{I}_l^{-1} & \mathbf{0}_n \end{pmatrix} + O(\eta^3), \tag{16}$$

where $\mathbf{0}_n$ is the matrix with zero entries. Hence, combining Eqs. (12) and (16), we find that

$$\rho(z) = \lim_{n\to\infty} \lim_{\eta\to 0} \frac{1}{\pi n} \partial^* \mathrm{Tr}_{21}\mathbf{H}^{-1}, \tag{17}$$

where $\mathrm{Tr}_{21}$ is a block trace over the diagonal of the lower-left block of $\mathbf{H}^{-1}$.

Defining the $jk$-th block of the generalized resolvent as

$$\mathsf{G}_{jk} = \begin{pmatrix} [\mathbf{H}^{-1}]_{j,k} & [\mathbf{H}^{-1}]_{j,k+n} \\ [\mathbf{H}^{-1}]_{j+n,k} & [\mathbf{H}^{-1}]_{j+n,k+n} \end{pmatrix}, \tag{18}$$

the spectral distribution (17) can be written as [37]

$$\rho(x,y) = \lim_{n\to\infty} \lim_{\eta\to 0} \frac{1}{\pi} \partial^* g_{21}, \tag{19}$$

where

$$\mathsf{g} = \begin{pmatrix} g_{11} & g_{12} \\ g_{21} & g_{22} \end{pmatrix} = \frac{1}{n}\sum_{j=1}^{n}\mathsf{G}_{jj}. \tag{20}$$

In Appendix A, we use the cavity method to derive a selfconsistent equation for the matrix g at fixed $\eta$ in the limit of $n \gg 1$, viz.,

$$\begin{pmatrix} g_{11} & g_{12} \\ g_{21} & g_{22} \end{pmatrix} = \left\langle \begin{pmatrix} \eta - \sigma^2 g_{22} & z - D - \tau\sigma^2 g_{21} \\ z^* - D - \tau\sigma^2 g_{12} & \eta - \sigma^2 g_{11} \end{pmatrix}^{-1} \right\rangle_D, \tag{21}$$

where $\langle\ldots\rangle_D$ denotes the average over the distribution $p_D$.

Note that to derive (21) we have determined g at finite values of $\eta$ in the limit of large $n$, and afterwards we take the limit of $\eta \to 0$. Hence, we interchange the two limits in Eq. (19), which is not evident as the leading order, correction terms in Eq. (16) at large values of $n$ and small values of $\eta$ intertwine the two limits. Demonstrating that these two limits can be interchanged constitutes the main challenge in rigorous approaches to non-Hermitian random matrix theory, see e.g. Refs. [31, 38–42]. This involves bounding the rate at which the least singular value of $z\mathbf{1}_n - \mathbf{A}$ converges to zero for large values of $n$, as the correction terms in (16) depend on the inverse of the matrices $\mathbf{I}_l$ and $\mathbf{I}_r$. In this paper, we use the theoretical physics approach, i.e., we exchange the two limits in good faith and then corroborate theoretical results with direct diagonalisation results. In the next section, we use the Eq. (21) together with Eq. (19) to determine the boundary of the support set of $\rho$ in the complex plane.

## 4 Boundary of the spectrum

The support set of $\rho(z)$ is defined as

$$\mathcal{S} = \overline{\{z \in \mathbb{C} : \rho(z) > 0\}}, \tag{22}$$

where $\overline{\cdot}$ denotes the closure of a set. From Eq. (10) it follows that the support set determines the leading eigenvalue whenever the spectrum does not contain eigenvalue outliers [27], which for the model defined in Sec. 2 is the case as long as $\mu = 0$.

The support set $\mathcal{S}$ follows from the solutions to the Eqs. (19)-(21). The Eq. (21) admits two types of solutions [17, 19]. First, there is the trivial solution for which $g_{11} = g_{22} = 0$ and $\partial_{z^*} g_{21} = 0$, yielding a distribution $\rho = 0$ for $z \notin \mathcal{S}$. Second, there is the nontrivial solution for which $g_{11} > 0$ and $g_{22} > 0$ and $\partial_{z^*} g_{21} \neq 0$, yielding the probability distribution $\rho > 0$ for $z \in \mathcal{S}$.

Although the trivial solution solves the set of Eqs. (21) for any value of $z$ and for $\eta = 0$, it is only for $z \notin \mathcal{S}$ that the trivial solution is relevant. Indeed, when $z \in \mathcal{S}$ the trivial solution is unstable with respect to infinitesimal small perturbations, and hence the regulator $\eta > 0$ guarantees that the spectral distribution for $z \in \mathcal{S}$ is determined by the nontrivial solution. As a consequence, the boundary of the support set $\mathcal{S}$ follows from a linear stability analysis of the Eqs. (21) around the trivial solution [19]. Expanding the Eqs. (21) in small values of $g_{11} > 0$ and $g_{22} > 0$, we obtain that for all values of $z \in \mathcal{S}$ it holds that

$$\left\langle \frac{\sigma^2}{(D - z^* + \tau\sigma^2 g_{12})(D - z + \tau\sigma^2 g_{21})} \right\rangle_D \geq 1, \tag{23}$$

and the boundary of the support set is given by

$$\left\langle \frac{\sigma^2}{(D - z^* + \tau\sigma^2 g_{12})(D - z + \tau\sigma^2 g_{21})} \right\rangle_D = 1. \tag{24}$$

Note that, in general, Eq. (24) is coupled with the Eq. (21) and therefore these equations have to be solved together.

In what follows, we first analyse the Eqs. (21)and (24) in two limiting cases, and then we discuss the general case.

### 4.1 Symmetric matrices with $J_{ij} = J_{ji}$ ($\tau = 1$)

For symmetric random matrices the Eq. (21) reduces to a functional equation for the resolvent of a Wigner matrix with diagonal disorder derived originally by Pastur in Ref. [24]. Indeed, in

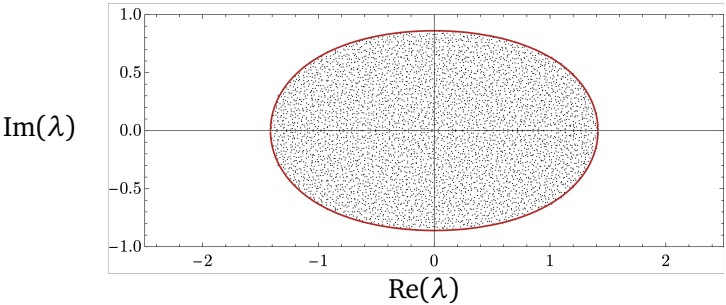

(a) Uniform distribution $p_D$ as defined in Eq. (7) with $d_+ = 1$ and $d_- = -1$; the parameter $\mu = 0$.

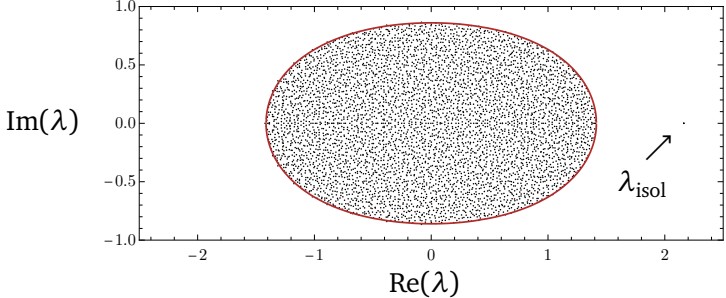

(b) Uniform distribution $p_D$ as defined in Eq. (7) with $d_+ = 1$ and $d_- = -1$; the parameter $\mu = 2$. The arrow points at the eigenvalue outlier

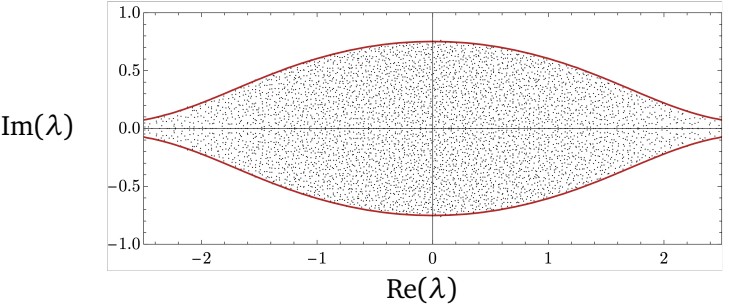

(c) Gaussian distribution $p_D$ as defined in Eq. (8); the parameter $\mu = 0$.

Figure 1: Spectra of three random matrices **A** as defined in Eq. (5) for the uncorrelated case $\tau = 0$ and with diagonal elements that are independently drawn from a uniform distribution [Panel (a) and Panel(b)] or a Gaussian distribution [Panel (c)]. Markers denote the eigenvalues of a random matrix of size $n = 3000$ and with off-diagonal elements $A_{ij} = J_{ij} + \mu/n$, where the $J_{ij}$ are drawn independently from a Gaussian distribution with zero mean and unit variance and where $\mu$ is as given in the subfigure captions. The red solid line denotes the solution to Eq. (27), which provides boundary of the support set $\mathcal{S}$ in the limit of infinitely large $n$. The eigenvalue outlier is indicated by an arrow in Panel (b). Panel (a) and (b) show the analytical solution Eq. (29) and Panel (c) is obtained by numerically solving Eq. (27).

this case $g_{22} = g_{11} = 0$ for all $z$ with nonzero imaginary part so that

$$g_{21} = \int_{\mathbb{R}} dx \, p_D(x) \frac{1}{z - x - \sigma^2 g_{21}} \, , \tag{25}$$

for all $z \notin \mathbb{R}$, which is identical to Equation (1.6) in Ref. [24]. Since $g_{21}$ is the Stieltjes transform of the spectral distribution defined on the real line, we can use the Sokhotski-Plemelj

inversion formula (see e.g. Chapter 8 of [43])

$$\rho(x + iy) = \frac{1}{\pi} \delta(y) \lim_{\epsilon \to 0^+} \text{Im}\left(g_{21}(x - i\epsilon)\right), \tag{26}$$

to obtain the spectral distribution. Note that the delta distribution $\delta(y)$ on the right hand-side of Eq. (26) specifies that the eigenvalues of $\mathbf{A}$ are real, and hence the distribution $\rho$ defined on the complex plane equals zero for all values $y \neq 0$.

## 4.2 Uncorrelated interaction variables $J_{ij}$ and $J_{ji}$ ($\tau = 0$)

In the absence of correlations between $J_{ij}$ and $J_{ji}$, the Eq. (24) decouples from the Eq. (21). Therefore, the "$\tau = 0$"-case is mathematically simpler to solve than the "$\tau \neq 0$"-case. The boundary of the support set $\mathcal{S}$ is determined by the values of $\lambda \in \mathbb{C}$ that solve the equation

$$1 = \sigma^2 \int_{\mathbb{R}} \mathrm{d}x \, p_D(x) \frac{1}{|\lambda - x|^2}, \tag{27}$$

which is closely related to the results obtained for the boundary of spectra of random directed graphs in Refs. [17, 18, 27] and to those of perturbed random matrices with uncorrelated matrix entries [28].

Equation (27) implies that for $\tau = 0$ the leading eigenvalue satisfies

$$\text{Re}(\lambda_1) \geq d_+ = \max\{x \in \mathbb{R} : p_D(x) > 0\}. \tag{28}$$

In other words, in the absence of correlations between the interaction variables $J_{ij}$ and $J_{ji}$, interactions always increase the real part of the leading eigenvalue and have thus a destabilising effect on system stability.

Let us analyse the boundary of the spectrum and the leading eigenvalue for a couple of examples. As shown in Appendix B, when $p_D(x)$ is the uniform distribution supported on the interval $[d_-, d_+]$, then the boundary of the support set $\mathcal{S}$ is given by values of $(x, y)$ that solve

$$\left((d_- - x)(d_+ - x) + y^2\right) = \frac{y(d_+ - d_-)}{\tan\left(\frac{y}{\sigma^2}(d_+ - d_-)\right)}, \quad y \in \left(-\frac{\pi\sigma^2}{d_+ - d_-}, \frac{\pi\sigma^2}{d_+ - d_-}\right) \setminus \{0\}, \tag{29}$$

a result that was also obtained in [29]. For $(d_+ - d_-)/\sigma^2 \ll 1$, we recover the celebrated circular law [31, 44], while for $(d_+ - d_-)/\sigma^2 \approx 1$ the formula Eq. (29) expresses a deformed circular law replacing the constant radius $\sigma$ by $y(d_+ - d_-)/\tan\left(\frac{y}{\sigma^2}(d_+ - d_-)\right)$. In Fig. 1(a) we have plotted the curve Eq. (29) for the case $d_+ = 1$ and $d_- = -1$ and we show that this theoretical results is well corroborated by the spectrum obtained from numerically diagonalising a matrix. From Eq. (29) it follows that the leading eigenvalue is given by

$$\text{Re}(\lambda_1) = \frac{1}{2}\left(\sqrt{(d_- - d_+)^2 + 4\sigma^2} + d_- + d_+\right). \tag{30}$$

Eq. (30) reveals that $\text{Re}(\lambda_1) > d_+$ for any value of $\sigma$, and hence the interactions make the system less stable. For $d_+ = d_- = d$ we recover the formula Eq. (4), and in the limit of large $d_-$ we get $\lim_{d_- \to -\infty} \text{Re}(\lambda_1) = d_+$.

As a second example we consider the case when $p_D$ is a Gaussian distribution with zero mean and unit variance. In Fig. 1(c), we compare the solution to Eq. (27) with the spectrum of a random matrix drawn from the ensemble defined in Sec. 2. In this case the spectrum $\mathcal{S}$ in the limit $n \to \infty$ contains the whole real axis, contrarily to the case where $p_D$ is a uniform distribution (compare Fig. 1(a) with Fig. 1(c)). The distinction between the two cases follows

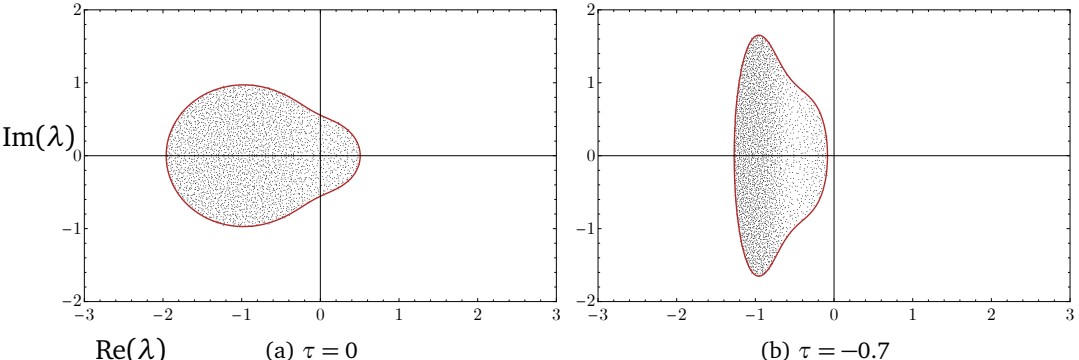

Figure 2: *Comparison between the spectra of two random matrices* **A** *with two different values of* $\tau$. Eigenvalues plotted are for two matrices of size $n = 3000$ whose diagonal elements are drawn from the bimodal distribution Eq. (6) with $d_- = -1$, $p = 0.9$ and $d_+ = 0.1$, and whose off-diagonal entries are drawn from a normal distribution with zero mean $\mu = 0$, variance $\sigma^2/n = 1/n$, and $\tau = 0$ [Panel (a)] or $\tau = -0.7$ [Panel (b)]. The red line denotes the solution to the Eqs. (32) and (33).

from the fact that $p_D$ is supported on a compact interval in the uniform case, while it is supported on the whole real axis in the Gaussian case. Indeed, Eq. (27) implies that in the former case the spectrum $\mathcal{S}$ is a finite subset of the complex plane, while in the latter case it contains the real axis. Consequently, for a compactly supported distribution $p_D$ the leading eigenvalue converges to a finite value as a function of $n$, while for a distribution $p_D$ that is supported on the real axis the leading eigenvalue diverges. The rate of divergence as a function of $n$ of the average of the leading eigenvalue, $\langle \lambda_1 \rangle$, is determined by the scaling of the maximum value of the diagonal entries $D_i$ as a function of $n$. Since the maximum of $n$ i.i.d. random variables drawn from a Gaussian distribution with zero mean and unit variance scales as $\sqrt{\log(n)}$ (see Theorem 1.5.3 in Ref. [45]), it holds that

$$\langle \lambda_1 \rangle = O_n(\langle D_{\max} \rangle) = O(\sqrt{\log(n)}), \tag{31}$$

when $p_D$ is Gaussian, where $D_{\max} = \max\{D_1, D_2, \ldots, D_n\}$ and where $O(\cdot)$ is the big O notation.

### 4.3 The case of generic correlations between $J_{ij}$ and $J_{ji}$ ($\tau \in [-1, 1]$)

We consider now the case of nonzero correlations between the interaction variables $J_{ij}$ and $J_{ji}$. In this case, it is more difficult to find the values of $z$ that solve the Eq. (24), as contrarily to the $\tau = 0$ case Eq. (24) is coupled with Eq. (21). Nevertheless, we can simplify the Eqs. (24) and (21) by using generic properties of **H** and **A**.

Using that **H** is Hermitian, which is implied by the definition Eq. (18), we obtain that $g_{12} = g_{21}^*$, $\text{Im}(g_{11}) = 0$ and $\text{Im}(g_{22}) = 0$. In addition, since **A** and $\mathbf{A}^T$ have the the same statistical properties, we can set $g_{11} = g_{22}$. Also, since we are interested in the boundary of the continuous part of the spectrum, which is located at the edge between the trivial and the nontrivial solutions, we can set $g_{11} = g_{22} = 0$, as this is satisfied for the trivial solution. Furthermore, we make the ansatz that $\text{Im}(g_{12})$ is independent of the distribution $p_D$, and therefore $\text{Im}(g_{12}) = y/\sigma^2(\tau - 1)$, which is the solution when $p_D(x) = \delta(x)$. In addition, using that $g_{11}g_{22} = 0$, we can express the Eq. (21) as

$$\text{Re}(g_{12}) = \left\langle \frac{D - x + \text{Re}(g_{12})\tau\sigma^2}{-((D - x) + \text{Re}(g_{12})\tau\sigma^2)^2 - \frac{y^2}{(1-\tau)^2}} \right\rangle_D, \tag{32}$$

and Eq. (24) reads

$$1 = \left\langle \frac{\sigma^2}{((D-x) + \mathrm{Re}(g_{12})\tau\sigma^2)^2 + \frac{y^2}{(1-\tau)^2}} \right\rangle_D . \tag{33}$$

We could not simplify these equations further, and hence we will obtain the boundary of the spectrum by solving the Eqs. (32-33).

In Figs. 2 and 3, we corroborate the boundary of the spectrum, obtained from solving the Eqs. (32-33), with numerical results for the eigenvalues of matrices of finite size, obtained with numerical diagonalisation routines. We show the boundary of the spectrum for the case of the bimodal distribution $p_D$ given by Eq. (6). Figure 2 compares two spectra with the same $\sigma$ but different values of $\tau$, whereas Fig. 3 considers one negative value of $\tau$ and observes how the spectrum changes as a function of $\sigma$. Note that the the real part of the leading eigenvalue $\mathrm{Re}(\lambda_1)$ decreases as a function of $\tau$.

The leading eigenvalue is obtained by solving Eqs. (32)-(33) at $y = 0$. For bimodal $p_D$ we obtain a quartic equation in $x$ and we identify the largest real-valued solution of this quartic equation with $\mathrm{Re}(\lambda_1)$. We have obtained an analytical expression for $\mathrm{Re}(\lambda_1)$ as a function of the system parameters, which we omit here as it is a long mathematical formula without clear use. However, it can be found in the Supplemental Material of Ref. [29].

For uniform $p_D$ the Eqs. (32)-(33) can be solved explicitly as shown in Appendix C. Remarkably, in this case we obtain a simple, analytical expression for the leading eigenvalue, viz.,

$$\begin{aligned} \mathrm{Re}(\lambda_1) &= \frac{1}{2}\left( \sqrt{(d_- - d_+)^2 + 4\sigma^2} + d_- + d_+ \right) \\ &+ \tau \frac{\sigma^2}{d_+ - d_-} \log\left( \frac{\sqrt{(d_- - d_+)^2 + 4\sigma^2} + d_+ - d_-}{\sqrt{(d_- - d_+)^2 + 4\sigma^2} - d_+ + d_-} \right), \end{aligned} \tag{34}$$

where $d_+ > d_- \in \mathbb{R}$. One readily verifies that for $\tau = 0$ Eq. (34) reduces to Eq. (30), for $\tau = 1$ Eq. (34) recovers the result in Ref. [26] for the case of symmetric matrices with entries drawn from a Gaussian distribution, and for $\sigma = 1$ it is equivalent to a formula that appeared in the Supplemental Material of [29]. Since the sign of the second term of Eq. (34) is equal to the sign of $\tau$, the leading eigenvalue $\lambda_1$ decreases as a function of negative values of $\tau$.

In Fig. 4 we compare Eq. (34) with numerical results of the leading eigenvalue obtained through the direct diagonalisation of matrices of finite size $n$. The numerics corroborate well the analytical results that are valid for infinitely large $n$. We make a few interesting observations from Fig. 4: (i) for $\tau = 0$, the leading eigenvalue is a monotonically increasing function of the interaction strength $\sigma$ implying a continuous increase of the width of the spectrum as a function of $\sigma$; (ii) for $\tau = -0.8$, the leading eigenvalue is a nonmonotonic function of $\sigma$. Initially, for small values of $\sigma$, the width of the spectrum decreases as a function of $\sigma$, while for large enough values of $\sigma$ the width of the spectrum increases as a function of $\sigma$; (iii) for $\tau = -1$, the leading eigenvalue is monotonically decreasing. In this case, the width of the spectrum decreases continuously as a function of $\sigma$ and converges for large $\sigma$ to a vertical spectrum centered on the mean value of $p_D$.

## 5 Eigenvalue outlier

Now, we determine the leading eigenvalue when $\mu \neq 0$. Even though, the continuous part of the spectrum is not affected by $\mu$ (see Appendix E), the spectrum may have an eigenvalue outlier, which can be the leading eigenvalue; this is illustrated in Panel (b) of Fig. 1. Hence,

for $\mu \neq 0$ the leading eigenvalue

$$\lambda_1 = \max\left\{\lambda_1^c, \lambda_{\text{isol}}\right\}, \tag{35}$$

(a) $\sigma = 0.5$

(b) $\sigma = 0.7$

(c) $\sigma = 1$

Figure 3: *Comparison between the spectra of random matrices* **A** *with different values of the interaction strength* $\sigma$. *Eigenvalues plotted are for three matrices of size* $n = 3000$ *whose off-diagonal elements* $(J_{ij}, J_{ji})$ *are drawn from a joint Gaussian distribution with zero mean, a Pearson correlation coefficient* $\tau = -0.7$, *and a variance* $\sigma^2/n$ *as indicated. The diagonal elements follow a bimodal distribution with parameters* $p = 0.9, d_- = -1, d_+ = 0.1$, *and* $\mu = 0$. *The red line denotes the solution to the Eqs. (32) and (33).*

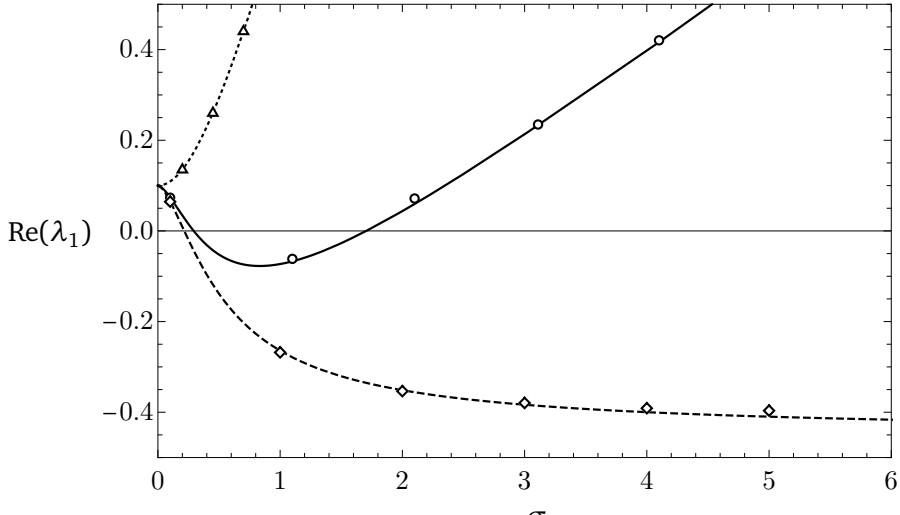

Figure 4: *Effect of the interaction strength $\sigma$ on the real part of the leading eigenvalue $\lambda_1$ when $\mu = 0$ and for $\tau = 0$ (triangle, dotted), $\tau = -0.8$ (circle, solid) and $\tau = -1$ (diamond, dashed).* Lines show the Eq. (34). Markers are numerical results obtained for random matrices **A** with diagonal elements $D_j$ that are drawn independently from a uniform $p_D$ supported on the interval $[d_-, d_+] = [-1, 0.1]$ and with pairs of off-diagonal elements $(J_{ij}, J_{ji})$ that are drawn independently from a normal distribution with mean 0, variance $\sigma^2/n$, and Pearson correlation coefficient $\tau$ as provided. Each marker represents the largest eigenvalue of one matrix realisation of size $n = 7000$.

where $\lambda_{\text{isol}}$ is the eigenvalue outlier, if it exists, and $\lambda_1^c$ is the leading eigenvalue of the continuous part of the spectrum, as defined by Eq. (10) with $\lambda_1$ replaced by $\lambda_1^c$. In what follows, we determine $\lambda_1^c$ and $\lambda_{\text{isol}}$.

The leading eigenvalue $\lambda_1^c$ of the support set $\mathcal{S}$ is, in the limit of large $n$, independent of $\mu$. Indeed, as shown in Appendix E, the boundary of the support set $\mathcal{S}$ solves the Eqs. (21) and (24), just as was the case for $\mu = 0$.

To determine the eigenvalue outlier we follow the theory for eigenvalue outliers of random matrices, as developed in Ref. [17], which is also based on the cavity method, albeit works in a different way as in this approach recursion relations are derived for entries of right eigenvectors, instead of the entries of the resolvent. Following this approach, we show in the Appendix F that the eigenvalue outliers $\lambda_{\text{isol}}$ of **A** in the limit for large $n$ solve

$$1 = \mu \, g_{21}(\lambda_{\text{isol}}), \quad \lambda_{\text{isol}} \notin \mathcal{S}, \tag{36}$$

where $g_{21}$ is the trivial solution to Eq. (21), i.e., for $g_{11} = g_{22} = 0$. For $\tau = 0$,

$$g_{21}(z) = \left\langle \frac{1}{z - D} \right\rangle_D, \tag{37}$$

which leads to the equation

$$1 = \mu \left\langle \frac{1}{\lambda_{\text{isol}} - D} \right\rangle_D, \tag{38}$$

for the outlier $\lambda_{\text{isol}}$.

In general for $\tau \neq 0$, we do not have an explicit expression for $g_{21}$, and hence we express $\lambda_{\text{isol}}$ in terms of the functional inverse $f_{21}$ of $g_{21}$, namely,

$$\lambda_{\text{isol}} = f_{21}\left(\frac{1}{\mu}\right), \tag{39}$$

where

$$z = f_{21}(g_{21}(z)). \tag{40}$$

Using Eq. (21) for $g_{22} = g_{11} = 0$, corresponding to the trivial solution, we obtain the following selfconsistent equation for $g_{21}(z)$,

$$g_{21} = \left\langle \frac{1}{z - D - \tau\sigma^2 g_{21}} \right\rangle_D, \quad \text{with} \quad z \notin \mathcal{S}. \tag{41}$$

If we can rewrite Eq. (41) as $z = f_{21}(g_{21})$, then we readily obtain the functional inverse $f_{21}$ of $g_{21}$.

Now, we determine $f_{21}$ for specific distributions of the diagonal disorder $D$. First, we consider the case when $p_D$ is uniform, as in Eq. (7). It then holds for real values of $z \in \mathbb{R}$ that $\text{Im}(g_{21}) = 0$, and

$$
\begin{aligned}
\text{Re}(g_{21}) &= \frac{1}{d_+ - d_-} \int_{d_-}^{d_+} du \frac{1}{z - u - \tau\sigma^2 \text{Re}(g_{21})} \\
&= \frac{1}{d_+ - d_-} \log \frac{z - d_- - \tau\sigma^2 \text{Re}(g_{21})}{z - d_+ - \tau\sigma^2 \text{Re}(g_{21})},
\end{aligned} \tag{42}
$$

from which it follows that

$$z = d_- + \frac{d_- + d_+}{e^{-(d_p - d_m)\text{Re}(g_{21})}} + \tau\sigma^2 \text{Re}(g_{21}), \tag{43}$$

and thus

$$f_{21}(u) = d_- + \frac{d_- + d_+}{e^{-(d_p - d_m)\text{Re}(u)}} + \tau\sigma^2 \text{Re}(u). \tag{44}$$

Inserting the expression Eq. (44) into Eq. (39), we obtain

$$\lambda_{\text{isol}} = d_- + \frac{d_- - d_+}{e^{-\frac{(d_+ - d_-)}{\mu}} - 1} + \tau\frac{\sigma^2}{\mu}, \tag{45}$$

which is an explicit analytical expression for the outlier when $D$ is uniformly distributed.

Equation (45) shows that for negative $\tau$ the eigenvalue outlier decreases monotonically as a function of $\sigma$, which is different from the nonmonotonic behaviour $\lambda_1^c$ as a function of $\sigma$, see Eq. (34). In Fig. 5, we plot $\lambda_1 = \max\{\lambda_1^c, \lambda_{\text{isol}}\}$ as a function of $\sigma$. For small values of $\sigma$, the leading eigenvalue $\lambda_1$ decreases rapidly as a function of $\sigma$, as $\lambda_1$ is an outlier, until $\lambda_{\text{isol}} = \lambda_1^c$, at which point the outlier stops existing and $\lambda_1$ is located at the boundary of $\mathcal{S}$.

Also for the case of bimodal $p_D$, as given by Eq. (6), we can obtain an explicit expression for $\lambda_{\text{isol}}$. Following the same steps as for uniform disorder, we get

$$\lambda_{\text{isol}} = \frac{1}{2}\left(d_- + d_+ + \mu + \sqrt{(d_+ - d_-)^2 + 2(d_- - d_+)(2p - 1)\mu + \mu^2}\right) + \tau\frac{\sigma^2}{\mu}. \tag{46}$$

Again, as in the case for uniform disorder, the outlier decreases monotonically as a function of $\sigma$ when $\tau < 0$.

Comparing Eqs. (45) with (46), we make the following interesting observation. Both equations take the form

$$\lambda_{\text{isol}} = \lambda_{\text{isol}}^{(0)} + \tau\frac{\sigma^2}{\mu}, \tag{47}$$

where $\lambda_{\text{isol}}^{(0)}$ is the corresponding eigenvalue outlier for $\tau = 0$ solving

$$1 = \mu\left\langle \frac{1}{\lambda_{\text{isol}}^{(0)} - D} \right\rangle_D. \tag{48}$$

Since Eq. (47) holds for both uniform and bimodal $p_D$, we conjecture that Eq. (47) holds for general $p_D$. The Eq. (47) is a convenient result as $\lambda_{\text{isol}}^{(0)}$ can be obtained easily from solving Eq. (48). Notice that for constant diagonal matrix entries, i.e., $D_i = d$ for all $i \in [n]$, Eq. (47) is consistent with Theorem 2.4 of Ref. [33] for the eigenvalue outliers of finite rank perturbations of elliptic random matrices.

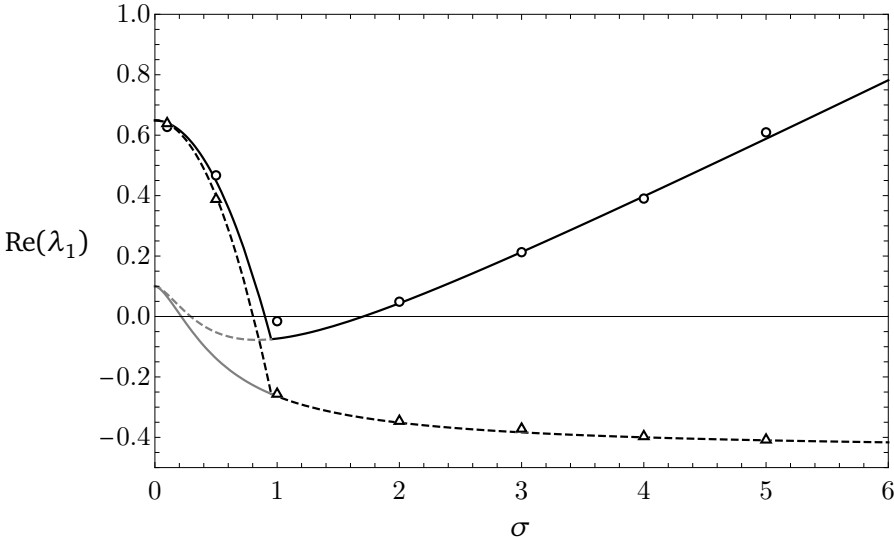

Figure 5: *Effect of the interaction strength $\sigma$ on the real part of the leading eigenvalue $\lambda_1$ for random matrices with $\mu = 2$ and all other parameters the same as in Fig. 4.* Similar to Fig. 4, solid lines/circles correspond with $\tau = -0.8$ and dashed lines/triangles with $\tau = -1$. Gray lines show Eq. (34). Black lines show the maximum between Eq. (45), for the eigenvalue outlier, and Eq. (34), for the leading eigenvalue of $\mathcal{S}$. Each marker represents the largest eigenvalue of one matrix realisation of size $n = 3000$.

## 6 Stability of linear dynamical systems

We discuss the implications of the spectral results obtained in the previous two sections for the stability of linear systems of the form given by Eq. (1).

### 6.1 Uncorrelated interactions destabilise dynamical systems

For $\tau = 0$ it holds that $\text{Re}(\lambda_1) \geq d_+$ for all values of $\sigma$ [see Eq. (28)], which has a couple of interesting implications for the stability of linear dynamical systems. First, a linear dynamical system with $\tau = 0$ cannot be stable if the support of $p_D$ covers the positive axis. Second, interactions $J_{ij}$ destabilise linear dynamical systems as $\lambda_1$ is an increasing function of $\sigma$ (see also Fig.3). Third, if the support of $p_D$ covers the whole real line, then the leading eigenvalue $\lambda_1$ diverges as a function of $n$. In the latter case we obtain a tradeoff between diversity, as measured by $n$, and stability, as measured by $\text{Re}(\lambda_1)$ [8, 46]. Indeed, when $p_D$ has unbounded support, then for any realisation of the system parameters $\sigma$, $\tau$, and $p_D$, there will exist a value $n^*$ so that with large probability $\text{Re}(\lambda_1) > 0$ when $n > n^*$. In Ref. [19], the latter scenario is referred to as size-dependent stability, as the system size $n$ is an important parameter in determining system stability.

## 6.2 Antagonistic interactions can render dynamical systems stable

In the case of negative $\tau$ values the interactions $J_{ij}$ can stabilise linear dynamical systems when they are neither too strong nor too weak. To understand how this works, consider linear systems $\mathbf{A}$ for which there exist values $x \in \mathbb{R}^+$ with $p_D(x) > 0$, such that the system is unstable in the absence of interactions. As illustrated in Fig. 3, adding antagonistic interactions to a linear system can retract the real part $\mathrm{Re}(\lambda_1)$ of the leading eigenvalue and make it negative. This example demonstrates that unlike the uncorrelated case with $\tau = 0$, interactions can contribute to the stability of a system when $\tau < 0$. However, as shown in Figs. 4 and 5, for large values of the interaction strength $\sigma$ the leading eigenvalue increases as a function of $\sigma$, and hence antagonistic interactions stabilise linear dynamical systems as long as they are neither too strong nor too weak.

Fig. 6 draws the lines of marginal stability, corresponding with $\mathrm{Re}(\lambda_1) = 0$, in the $(\sigma, \tau)$ plane for $\mu \leq 0$ and for homogeneous growth rates $p_D(x) = \delta(x - D)$ (dotted line), for a bimodal distributions $p_D$ (dashed line), and for a uniform distribution $p_D$ (solid line). In these cases, the leading eigenvalue $\lambda_1$ is located at the boundary of the support set $\mathcal{S}$, such that, $\lambda_1 = \lambda_1^c$. For all cases we have set $\langle D \rangle = -1$, so that we see the effect of fluctuations in $D$ on system stability. Note that for the dotted line $d_+ = -1$, whereas for the dashed and solid lines $d_+ = 0.1$. As a consequence, for the dotted line a stable region exists when $\tau = 0$ and $\sigma$ is small enough, while for the other cases there is no stable region when $\tau = 0$. Interestingly, for negative values of $\tau$ and for interaction strengths $\sigma$ that are neither too weak nor too strong, there exists a stable region with $\mathrm{Re}(\lambda_1) < 0$. This region exists even though $d_+ > 0$ (solid and dashed lines). On the other hand, for $\tau = 0$ a stable region can only exist when $d_+ < 0$, which is the case of the dotted line with homogeneous rates.

Figure 6 shows that $\mathrm{Re}(\lambda_1)$ is independent of $p_D$ for large values of $\sigma$ and fixed $\langle D \rangle$. We explore this universal behaviour in more depth. Expanding the expression of $\mathrm{Re}(\lambda_1)$ for Eq. (34) in large values of $\sigma$ we obtain

$$\mathrm{Re}(\lambda_1) = (1 + \tau)\sigma + \frac{1}{2}(d_- + d_+) + (3 + \tau)\frac{(d_- - d_+)^2}{24\sigma} + O(1/\sigma^2). \tag{49}$$

Identifying the mean and variance of the uniform distribution $p_D$ in Eq. (49), we can write

$$\mathrm{Re}(\lambda_1) = (1 + \tau)\sigma + \langle D \rangle + \frac{(3 + \tau)}{2\sigma}\langle\langle D^2 \rangle\rangle + O(1/\sigma^2), \tag{50}$$

where $\langle\langle D^2 \rangle\rangle$ represents the variance of the diagonal elements. If $D$ is a deterministic variable with zero variance, then we recover the Eq. (4). This suggests that when the interactions are strong enough, only the first moment of the diagonal elements is important, rather than the distribution of their elements. Although the relation Eq. (50) is derived for the uniform case, numerical evidence shows that it also holds for the bimodal case, and therefore we conjecture that it holds for arbitrary $p_D$ distributions. Demonstrating the validity of the Eq. (50) beyond the uniform case would be an interesting extension of the present work.

Figure 7 draws the lines of marginal stability in the $(\sigma, \tau)$, similar to Fig. 6, albeit with $\mu > 0$. In this case, the leading eigenvalue $\lambda_1$ is either the eigenvalue outlier, $\lambda_1 = \lambda_{\mathrm{isol}}$, when it exists, or is the leading eigenvalue of the support set $\mathcal{S}$, i.e., $\lambda_1 = \lambda_1^c$. The eigenvalue outlier exists when $\sigma$ is small enough, and in this regime the system stability increases as a function of $\sigma$. For fixed $\tau$, at a value $\sigma = \sigma^*$ the eigenvalue outlier merges into $\mathcal{S}$, i.e., $\lambda_{\mathrm{isol}} = \lambda_1^c$, and for $\sigma > \sigma^*$ the leading eigenvalue is $\lambda_1^c$. We can clearly notice this transition in Fig. 7 due to the cusp that appears in the lines of marginal stability. Hence, for $\sigma > \sigma^*$, the lines of marginal stability in Fig. 7 are identical to those in Fig. 6. Comparing Panels (a) and (b) in Fig. 7, we notice that increasing the parameter $\mu$ reduces system stability. This can be understood from the fact that the eigenvalue outlier $\lambda_{\mathrm{isol}}$ increases as a function of $\mu$, while the boundary of the

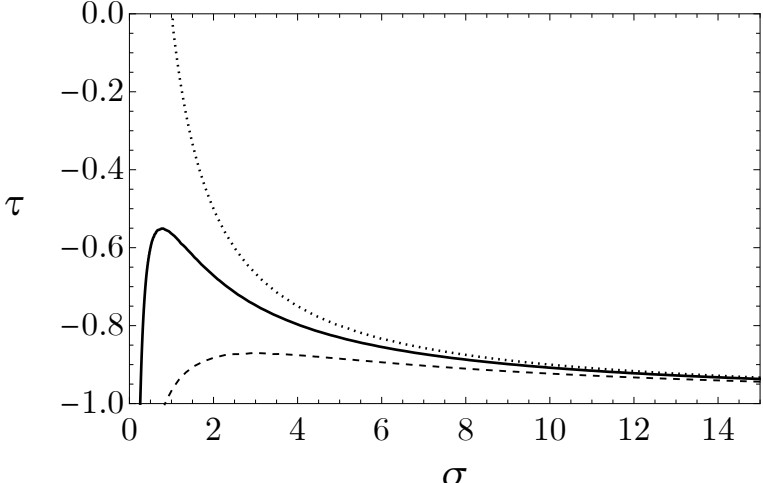

Figure 6: *Phase diagram for the stability of linear dynamical systems with antagonistic interactions when $\mu \leq 0$. Lines denote values of $(\tau, \sigma)$ of marginal stability, i.e. $\text{Re}(\lambda_1) = 0$, separating a stable region with $\text{Re}(\lambda_1) < 0$ (below the lines) from an unstable region with $\text{Re}(\lambda_1) > 0$ (above the lines). Results shown are for the random matrix model defined in Sec. 2 and for various distributions $p_D$ with the same mean $\langle D \rangle = -1$. The solid line represents a uniform disorder on the interval $I = [-2.1, 0.1]$; the dashed line represents a bimodal disorder with parameters $p = 0.5, d_- = -2.1, d_+ = 0.1$; and the dotted line represents the case where all diagonal elements take the value $-1$ with no disorder.*

continuous spectrum is independent of $\mu$. Note that negative values of $\mu$ have no influence on system stability as in this case the eigenvalue outlier, when it exists, is located on the real axis at the negative side of $\mathcal{S}$.

## 7 Discussion

We have obtained exact results for the leading eigenvalue of random matrices of the form Eq. (5), where the pairs $(J_{ij}, J_{ji})$ are i.i.d. random variables drawn from a joint distribution with moments as given in Eq. (3), and where the diagonal elements $D_i$ are i.i.d. random variables drawn from a distribution $p_D$.

If the Pearson correlation coefficient $\tau = 0$, then the boundary of the spectrum solves the Eq. (27), which implies that $\lambda_1 \geq d_+$ [see Eq. (28)]. Hence, in this case interactions render dynamical systems less stable, irrespective of the form of $p_{J_1, J_2}$ and $p_D$. On the other hand, if the Pearson correlation coefficient $\tau$ between the pairs $(J_{ij}, J_{ji})$ is negative and the variance of the distribution $p_D$ is nonzero, then $\lambda_1$ can exhibit a nonmonotonic behaviour as a function of the strength $\sigma$ of the off-diagonal matrix elements, as illustrated in Figs. 4 and 5. As a consequence, antagonistic interactions that are neither too strong nor too weak can stabilise linear dynamical systems when the diagonal entries $D_i$ are heterogeneous, see Fig. 6.

The results in Figs. 4 and 5 can also be understood perturbatively. Indeed, a perturbative expansion of the eigenvalues of $\mathbf{A}$ in the parameter $\sigma$ starting from the diagonal case with $\sigma = 0$ leads to the expression (see Appendix D)

$$\lambda_j(\sigma) = D_j + \sum_{i=1;(i \neq j)}^{n} \frac{J_{ij} J_{ji}}{(D_j - D_i)} + O(\sigma^3). \tag{51}$$

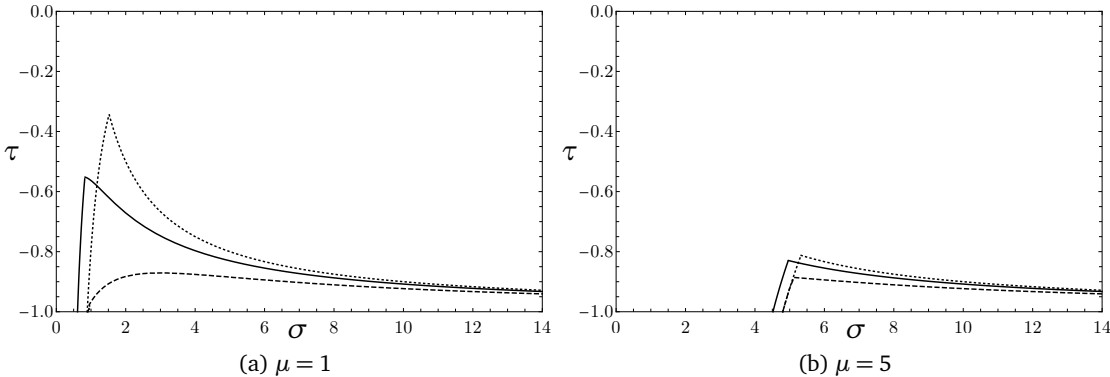

Figure 7: *Phase diagram for the stability of linear dynamical systems with antagonistic interactions and for $\mu > 0$. Lines show parameter values for which the system is marginally stable ($\text{Re}[\lambda_1] = 0$). The parameters are the same as in Fig. 6, except for $\mu$, which is set to $\mu = 1$ in Panel (a) and $\mu = 5$ in Panel (b).*

For the leading eigenvalue $j = 1$ it holds that $D_1 - D_i \geq 0$ for all values $i \in [n]$, and hence the second term is negative whenever $J_{ij}J_{ji} < 0$, leading to the initial decrease of the leading eigenvalue $\lambda_1$ in Fig. 4. For larger values of $\sigma$ we need to consider the higher order terms in the perturbative expansion of $\lambda_1$, which are in general positive leading to the nonmonotonic behaviour of $\lambda_1$ in Fig. 4.

We discuss various interesting questions for future research, both from the mathematical and ecological point of view.

For distributions $p_D$ that are uniform and bimodal, we have obtained the analytical expressions Eqs. (46) and (45) for the eigenvalue outlier, which led us to conjecture Eq. (47) for general distributions $p_D$. However, we have no proof of this general expression for the eigenvalue outlier, and hence it will be interesting to construct a proof of this simple, generic formula in future work.

For the case of a uniform distribution $p_D$ of the diagonal elements we have obtained an analytical expression for $\lambda_1$, which is given either by Eq. (34) or Eq. (45), depending on the value of $\mu$, and in the case $\tau = 0$ we have obtained a closed form expression for the boundary of the support of the spectral density $\rho$, which is given by Eq. (29) and which also holds for uniform $p_D$. The peculiar solvability of the uniform disorder case is consistent with results obtained recently in Ref. [26] for symmetric matrices ($\tau = 1$). Reference [26] shows that when $D_j = a + bj/n$, with $a$ and $b$ arbitrary constants, and when the entries $J_{ij}$ are complex-valued and Gaussian distributed, then an explicit expression for the joint distribution of eigenvalues can be obtained. Based on the results in the present paper, one may speculate that these results are extendable to the case of $\tau = 0$, which will be interesting to explore in future work.

An important extension of the present work are models of the form

$$A_{ij} = (J_{ij} + \mu)C_{ij} + \delta_{i,j}D_i, \tag{52}$$

where $C_{ij}$ is now the adjacency matrix of a random graph. The case of random directed graphs with a prescribed degree distribution $p_{K^{\text{in}},K^{\text{out}}}$ of indegrees and outdegrees has been solved in Refs. [17, 18, 27]. This is the sparse equivalent of the "$\tau = 0$"-case, and in fact the oriented and locally tree-like structure of random directed graphs leads to a decoupling similar to those of Eqs. (21) and (24) in the "$\tau = 0$"-case. For this reason, random directed graphs are analytically tractable, and Refs. [17, 18, 27] derived for the boundary of the spectrum an equation similar

to Eq. (27), but with a prefactor that is given by $(\sigma^2 + \mu^2)\frac{\langle K^{\text{in}}K^{\text{out}}\rangle}{c}$, i.e.,

$$1 = (\sigma^2 + \mu^2)\frac{\langle K^{\text{in}}K^{\text{out}}\rangle}{c}\int_{\mathbb{R}} dx\, p_D(x)\frac{1}{|\lambda - x|^2}\,, \tag{53}$$

and analogously, Refs. [17, 18, 27] derived for the eigenvalue outliers an equation similar to Eq. (54), viz.,

$$1 = \mu\frac{\langle K^{\text{in}}K^{\text{out}}\rangle}{c}\int_{\mathbb{R}} dx\, p_D(x)\frac{1}{\lambda_{\text{isol}} - x}\,. \tag{54}$$

The case of antagonistic interactions ($\tau < 0$) is considerably more challenging as one needs to know the distribution of the diagonal entries $[\mathbf{G}]_{ii}$ of the resolvent, which is non trivial in the sparse case, see Ref. [19]. Nevertheless, Ref. [19] analysed the antagonistic case without diagonal disorder and found that systems with antagonistic interactions are significantly more stable than systems with mutualistic and competitive interactions (in fact, in the limit $n \to \infty$ they are infinitely more stable). Ref. [19] did however not study the effect of diagonal disorder on system stability.

In an ecological setting, the Jacobian matrix of a set of randomly coupled Lotka-Volterra equations have a specific structure, viz., all elements in a row are multiplied by the population abundance so that

$$A_{ij} = D_i J_{ij}\,. \tag{55}$$

The spectra of such random matrix ensembles have been studied in Refs. [48,49], and it would be interesting to study the stabilising effect of antagonistic interactions in this setup.

The question of stability is also relevant for the study of experimental systems, see e.g. Refs. [50–52], and the matrices studied in the present paper are null models for real-world systems. However, as discussed in detail in Ref. [53], most ecological data on foodwebs is qualitative, and obtaining quantative data in particular on the Jacobian, is challenging.

Other interesting applications of the theory developed in the present paper are the study of Turing patterns that are governed by randomly coupled chemical reactions, which in Fourier space involves a random matrix with diagonal disorder [54].

Let us end the paper with a word of caution when using the present results to understand the dynamics of nonlinear systems. In a linear system there exist one fixed point, i.e., $\vec{x} = 0$, and the system parameters will not affect the existence and uniqueness of this fixed point. However, in nonlinear systems this is in general not the case, see e.g. [55, 56], and therefore system stability can also be affected by bifurcations that eliminate fixed points. Moreover, for certain problems, such as stability in ecosystems, the fixed point has to be feasible, which for ecosystems implies that all entries of the fixed point are nonnegative [1], and this also constitutes an interesting random matrix theory problem [57]. Another issue is that $\mathbf{A}$ is the Jacobian matrix, which is in general different from the interaction matrix. Nevertheless, studies in, among others, ecology [12–14] and neuroscience [2,3], show that nonlinear systems do exhibit regimes with one unique stationary fixed point and random matrix theory can provide insights on system stability in this regime. In the ecological context for symmetric interactions $J_{ij} = J_{ji}$, Refs. [12] shows that May's stability argument, albeit in the symmetric setting, applies when the number of extinct species is correctly taking into account, and the corresponding spectrum of the Hessian is described by random matrix theory. Moreover, note that for symmetric interactions replica theory can be used to determine the leading eigenvalue of the Hessian, see Refs. [12,14,58], while for nonsymmetric interactions this is not possible.

When preparing the manuscript, we became aware of the preprint [59] that also studies the spectral properties of matrices of the type defined in Sec. 2. However, the paper [59] discusses the case of $\tau > 0$ for which interactions further destabilise fixed points, whereas we were interested in the potentially stabilising effect of interactions for $\tau < 0$.

## Acknowledgements

IN thanks Jean-Philippe Bouchaud for several useful email communications, and Andrea Mambuca, Chiara Cammarota, and Pietro Valigi for fruitful discussions. SC thanks Joseph Baron for pointing out Reference [29], and SC thanks Giorgio Carugno for a fruitful discussion.

## A  Derivation of the generalised resolvent equation (21) using the Schur complement formula

We derive the Eq. (21) using two useful properties.

First, we use that permutation of a matrix and matrix inversion are two commutable operations. Indeed, let $\mathbf{P}$ be the orthogonal matrix that represents an arbitrary permutation of the integers $\{1, 2, \ldots, n\}$, then

$$\mathbf{P}\mathbf{H}^{-1}\mathbf{P}^{-1} = \left(\mathbf{P}\mathbf{H}\mathbf{P}^{-1}\right)^{-1}. \tag{56}$$

We use this property to perform the permutation [18,37]

$$[\mathbf{H}]_{j,k} \rightarrow \begin{cases} \left[\tilde{\mathbf{H}}\right]_{2j-1,2k-1}, & \text{if} & 1 \le j,k \le n, \\ \left[\tilde{\mathbf{H}}\right]_{2j-n,2k-1}, & \text{if} & n+1 \le j \le 2n,\ 1 \le k \le n, \\ \left[\tilde{\mathbf{H}}\right]_{2j-1,2k-n}, & \text{if} & 1 \le j \le n,\ n+1 \le k \le 2n, \\ \left[\tilde{\mathbf{H}}\right]_{2j-n,2k-n}, & \text{if} & n+1 \le j,k \le 2n, \end{cases} \tag{57}$$

where $\tilde{\mathbf{H}}$ is the matrix of permuted entries of $\mathbf{H}$. The effect of this permutation is to bundle together the elements of $\mathbf{H}$ that depend on pairs of entries $(A_{ij}, A_{ji})$.

Second, we use the Schur inversion formula for the inverse of a $2 \times 2$ block matrix [37,60],

$$\begin{pmatrix} \mathbf{a} & \mathbf{b} \\ \mathbf{c} & \mathbf{d} \end{pmatrix} = \begin{pmatrix} \mathbf{s_d}^{-1} & -\mathbf{s_d}\mathbf{b}\mathbf{d}^{-1} \\ -\mathbf{d}^{-1}\mathbf{c}\mathbf{s_d} & \mathbf{s_a}^{-1} \end{pmatrix}, \tag{58}$$

where $\mathbf{s_d} = \mathbf{a} - \mathbf{b}\mathbf{d}^{-1}\mathbf{c}$ and $\mathbf{s_a} = \mathbf{d} - \mathbf{c}\mathbf{a}^{-1}\mathbf{b}$ are the Schur complements of the blocks $\mathbf{d}$ and $\mathbf{a}$, respectively. If we choose for $\mathbf{a}$ the upper diagonal $2 \times 2$ block of $\tilde{\mathbf{H}}$, then we obtain

$$\mathsf{G}_{11} = \left(\begin{pmatrix} \eta & z - D_1 \\ z^* - D_1 & \eta \end{pmatrix} - \frac{1}{n}\sum_{j=2}^{n}\sum_{\ell=2}^{n}\begin{pmatrix} 0 & J_{1j} \\ J_{j1} & 0 \end{pmatrix}\mathsf{G}_{j\ell}^{(1)}\begin{pmatrix} 0 & J_{\ell 1} \\ J_{1\ell} & 0 \end{pmatrix}\right)^{-1}, \tag{59}$$

where $\mathsf{G}_{j\ell}^{(1)}$ is defined as in Eq. (18), but for a matrix $\mathbf{A}^{(1)}$ obtained by deleting the 1-th row and 1-th column of $\mathbf{A}$. Permuting the entries of the matrix so that the 1-th row and 1-th column are swapped with the $i$-th row and $i$-th column, and using again Eq. (56), we obtain the analogous formula

$$\mathsf{G}_{ii} = \left(\begin{pmatrix} \eta & z - D_i \\ z^* - D_i & \eta \end{pmatrix} - \frac{1}{n}\sum_{j=1; j \neq i}^{n}\sum_{\ell=1; \ell \neq i}^{n}\begin{pmatrix} 0 & J_{ij} \\ J_{ji} & 0 \end{pmatrix}\mathsf{G}_{j\ell}^{(i)}\begin{pmatrix} 0 & J_{\ell i} \\ J_{i\ell} & 0 \end{pmatrix}\right)^{-1}. \tag{60}$$

Note that the sum in Eq. (60) that runs over the indices $\ell$ and $j$ contains a very large number of terms in the limit of $n \gg 1$. Assuming that the law of large numbers applies to this sum — which can be verified to be the case, see Sec. 3 of Ref. [30] — we replace the sum by

its average value leading to

$$\sum_{j=1;j\neq i}^{n}\sum_{\ell=1;\ell\neq i}^{n}\begin{pmatrix} 0 & J_{ij} \\ J_{ji} & 0 \end{pmatrix}\mathsf{G}_{j\ell}^{(i)}\begin{pmatrix} 0 & J_{\ell i} \\ J_{i\ell} & 0 \end{pmatrix}$$

$$= (n-1)\begin{pmatrix} \langle J_{iu}^2\rangle\langle[\mathsf{G}_{uu}^{(i)}]_{22}\rangle & \langle J_{iu}J_{ui}\rangle\langle[\mathsf{G}_{uu}^{(i)}]_{21}\rangle \\ \langle J_{iu}J_{ui}\rangle\langle[\mathsf{G}_{uu}^{(i)}]_{12}\rangle & \langle J_{ui}^2\rangle\langle[\mathsf{G}_{uu}^{(i)}]_{11}\rangle \end{pmatrix}$$

$$+(n-1)(n-2)\begin{pmatrix} \langle J_{iu}\rangle\langle J_{iv}\rangle\langle[\mathsf{G}_{uv}^{(i)}]_{22}\rangle & \langle J_{iu}\rangle\langle J_{vi}\rangle\langle[\mathsf{G}_{uv}^{(i)}]_{21}\rangle \\ \langle J_{iv}\rangle\langle J_{ui}\rangle\langle[\mathsf{G}_{uv}^{(i)}]_{12}\rangle & \langle J_{ui}\rangle\langle J_{vi}\rangle\langle[\mathsf{G}_{uv}^{(i)}]_{11}\rangle \end{pmatrix},\tag{61}$$

with $u,v\in[n]\setminus\{i\}$ and $u\neq v$. In Eq. (61) we have used that the pairs $(J_{ij},J_{ji})$ are identically and independently distributed random variables. Moreover, we have used that $\langle[\mathsf{G}_{uu}^{(i)}]_{12}\rangle$ and $\langle[\mathsf{G}_{uv}^{(i)}]_{12}\rangle$ are independent of $i$, $u$ and $v$, as in the definition of the random matrix model for $\mathbf{A}$ all indices are equivalent. Since $\langle J_{iu}\rangle=0$, the second term in Eq. (61) equals zero, and using that $\sigma^2=\langle J_{ik}^2\rangle$ and $\tau\sigma^2=\langle J_{ik}J_{ki}\rangle$, we obtain

$$\mathsf{G}_{ii}=\left(\begin{pmatrix} \eta & z-D_i \\ z^*-D_i & \eta \end{pmatrix}-\sigma^2\begin{pmatrix} \langle[\mathsf{G}_{uu}^{(i)}]_{22}\rangle & \tau\langle[\mathsf{G}_{uu}^{(i)}]_{21}\rangle \\ \tau\langle[\mathsf{G}_{uu}^{(i)}]_{12}\rangle & \langle[\mathsf{G}_{uu}^{(i)}]_{11}\rangle \end{pmatrix}\right)^{-1}+o_n(1),\tag{62}$$

where $o(\cdot)$ is the small o notation. Taking the ensemble average of Eq. (62) and, in the limit of large $n$, identifying

$$g=\langle\mathsf{G}_{ii}\rangle,\tag{63}$$

for all $i\in[n]$, and

$$g=\langle\mathsf{G}_{uu}^{(i)}\rangle,\tag{64}$$

for all $u\in[n]\setminus\{i\}$, we obtain Eq. (21). Equation (64) follows from the fact that $\mathbf{A}^{(i)}$ is drawn from the same ensemble as $\mathbf{A}$, except that $n\to n-1$.

## B  Derivation of Eq. (29) for the boundary of the spectrum when $\tau=0$ and $p_D$ is uniform

Equation (27) for the uniform distribution Eq. (7) gives

$$\int_{d_-}^{d_+}\frac{1}{(x-u)^2+y^2}\mathrm{d}u=\frac{d_+-d_-}{\sigma^2}\,,\tag{65}$$

where we have used $z=x+\mathrm{i}y$. Using the formula

$$\int\frac{1}{x^2+a^2}dx=\frac{1}{a}\arctan\frac{x}{a}+\text{constant}\,,\tag{66}$$

for the indefinite integral of $1/(x^2+a^2)$ when $a\neq0$, we obtain that for $y\neq0$

$$\arctan\left[\frac{(d_+-x)}{y}\right]-\arctan\left[\frac{(d_--x)}{y}\right]=\frac{(d_+-d_-)}{\sigma^2}y\,.\tag{67}$$

Notice that since $\arctan(a)\in(-\pi/2,\pi/2)$, we have the condition $y\in(-\pi\sigma^2/(d_+-d_-),\pi\sigma^2/(d_+-d_-)$. Subsequently, using

$$\arctan a-\arctan b=\arctan\left(\frac{a-b}{1+ab}\right)\tag{68}$$

in Eq. (67), we obtain Eq. (29).

## C Derivation of Eq. (34) for the leading eigenvalue $\lambda_1$ in the case of uniformly distributed diagonal elements

We derive Eq. (34) for the leading eigenvalue $\lambda_1$ when $p_D$ is the uniform distribution given by Eq. (7).

Using the assumption that the leading eigenvalue $\lambda_1 \in \mathbb{R}$, we set $y = 0$ in equations Eqs. (32-33) yielding

$$\text{Re}(g_{12}) = -\left\langle \frac{1}{(D-x) + \text{Re}(g_{12})\tau\sigma^2} \right\rangle_D , \tag{69}$$

and

$$1 = \left\langle \frac{\sigma^2}{[(D-x) + \text{Re}(g_{12})\tau\sigma^2]^2} \right\rangle_D . \tag{70}$$

Integrating the equations Eq. (69-70) over the uniform distribution $p_D$ supported on the interval $[d_-, d_+]$, we obtain

$$\text{Re}(g_{12}) = \frac{\log\left(x - d_- - \text{Re}(g_{12})\sigma^2\tau\right)}{d_+ - d_-} - \frac{\log\left(x - d_+ - \text{Re}(g_{12})\sigma^2\tau\right)}{d_+ - d_-} , \tag{71}$$

and

$$1 = \frac{\sigma^2}{(d_+ - x + \text{Re}(g_{12})\tau\sigma^2)(d_- - x + \text{Re}(g_{12})\tau\sigma^2)} . \tag{72}$$

We first solve Eq. (72) towards $\text{Re}(g_{12})$ with solutions

$$\text{Re}(g_{12}) = \frac{x}{\sigma^2\tau} - \frac{(d_- + d_+)}{2\sigma^2\tau} + s\frac{\sqrt{(d_+ - d_-)^2 + 4\sigma^2}}{2\sigma^2\tau} , \tag{73}$$

where $s = \pm 1$. Replacing $\text{Re}(g_{12})$ in Eq. (71) by this solution gives a linear equation in $x$. The solutions of this linear equation provide the intersection points of the boundary of the support of the spectral distribution with the real axis, viz.,

$$\begin{aligned} x &= \frac{1}{2}\left(-s\sqrt{(d_- - d_+)^2 + 4\sigma^2} + d_- + d_+\right) \\ &\quad + \tau\frac{\sigma^2}{d_+ - d_-}\log\left(\frac{-s\sqrt{(d_- - d_+)^2 + 4\sigma^2} + d_+ - d_-}{-s\sqrt{(d_- - d_+)^2 + 4\sigma^2} - d_+ + d_-}\right) . \end{aligned} \tag{74}$$

For $s = 1$ we obtain the leading eigenvalue given by Eq. (34).

## D Perturbation theory for the leading eigenvalue

We use perturbation theory to understand the functional behaviour in Fig. 4 of $\text{Re}(\lambda_1)$ as a function of $\sigma$.

Let $\mathbf{D} + \sigma\mathbf{J}$, where $\mathbf{D}$ is a diagonal matrix, $\sigma$ a small parameter, and $\mathbf{J}$ an arbitrary $\sigma$-independent matrix with zero-valued diagonal entries. Let $\lambda_j^{(0)}$, $\vec{r}_j^{(0)}$, and $\vec{l}_j^{(0)}$ denote the eigenvalues, right eigenvectors, and left eigenvectors of $\mathbf{D}$, respectively.

Let $\lambda_j(\sigma)$ denote the eigenvalues of $\mathbf{D} + \sigma\mathbf{J}$. An expansion around $\sigma \approx 0$ gives

$$\lambda_j(\sigma) = \lambda_j^{(0)} + \lambda_j^{(1)}\sigma + \lambda_j^{(2)}\sigma^2 + O(\sigma^3), \tag{75}$$

with $\lambda_j^{(0)} = D_j$. Note that in this paper we use the convention that $\lambda_1^{(0)} \geq \lambda_2^{(0)} \geq \ldots \lambda_n^{(0)}$ and thus $D_1 \geq D_2 \geq \ldots D_n$.

It then holds [61]

$$\lambda_j^{(1)} = \frac{\vec{l}_j^{(0)} \cdot \mathbf{J} \vec{r}_j^{(0)}}{\vec{l}_j^{(0)} \cdot \vec{r}_j^{(0)}}, \tag{76}$$

and

$$\lambda_j^{(2)} = \frac{1}{\vec{l}_j^{(0)} \cdot \vec{r}_j^{(0)}} \sum_{i=1;i\neq j}^{n} \frac{[\vec{l}_j^{(0)} \cdot \mathbf{J} \vec{r}_i^{(0)}][\vec{l}_i^{(0)} \cdot \mathbf{J} \vec{r}_j^{(0)}]}{(\vec{l}_i^{(0)} \cdot \vec{r}_i^{(0)})(\lambda_j^{(0)} - \lambda_i^{(0)})}. \tag{77}$$

Since $\mathbf{D}$ is a diagonal matrix, we can set $\vec{l}_i^{(0)} \cdot \vec{r}_i^{(0)} = \delta_{i,j}$ and $\vec{l}_j^{(0)} \cdot \mathbf{J} \vec{r}_i^{(0)} = J_{ji}(1 - \delta_{i,j})$. In this case, it holds that

$$\lambda_j(\sigma) = D_j + \sum_{i=1;i\neq j}^{n} \frac{J_{ij} J_{ji}}{(D_j - D_i)} \sigma^2 + O(\sigma^3). \tag{78}$$

For $\sigma = 0$, it holds that $D_1 = D_{\max}$ and thus the denonimator in Eq. (78) is positive. From this it follows that when $J_{ij} J_{ji} < 0$, $\lambda_1$ initially decreases as a function of $\sigma$. However, when $\sigma$ is larger, then the $O(\sigma^3)$ becomes relevant, which provides the nonmonotonic behaviour in Fig. 6.

# E Boundary of the spectrum when $\mu \neq 0$

We show that the support set $\mathcal{S}$ of $\rho(z)$ is, in the limit of large $n$, independent of $\mu$, and hence in this limit the boundary of $\mathcal{S}$ solves the Eqs. (21) and (24).

Following the derivation of Appendix A, we obtain, instead of the self-consistent Eq. (60) for $\mathsf{G}_{ii}$, the self-consistent equation

$$
\begin{aligned}
\mathsf{G}_{ii} \;=\; & \left( \begin{pmatrix} \eta & z - D_i + \mu/n \\ z^* - D_i + \mu/n & \eta \end{pmatrix} - \right. \\
& \left. \frac{1}{n} \sum_{j=1;j\neq i}^{n} \sum_{\ell=1;\ell\neq i}^{n} \begin{pmatrix} 0 & J_{ij} + \mu/n \\ J_{ji} + \mu/n & 0 \end{pmatrix} \mathsf{G}_{j\ell}^{(i)} \begin{pmatrix} 0 & J_{\ell i} + \mu/n \\ J_{i\ell} + \mu/n & 0 \end{pmatrix} \right)^{-1}.
\end{aligned}
\tag{79}
$$

Assuming the law of large numbers applies, we obtain for the second term in the previous equation,

$$
\begin{aligned}
\sum_{j=1;j\neq i}^{n} \sum_{\ell=1;\ell\neq i}^{n} & \begin{pmatrix} 0 & J_{ij} + \mu/n \\ J_{ji} + \mu/n & 0 \end{pmatrix} \mathsf{G}_{j\ell}^{(i)} \begin{pmatrix} 0 & J_{\ell i} + \mu/n \\ J_{i\ell} + \mu/n & 0 \end{pmatrix} \\
= \; (n-1) & \begin{pmatrix} \left( \langle J_{iu}^2 \rangle + \frac{\mu^2}{n^2} \right) \langle [\mathsf{G}_{uu}^{(i)}]_{22} \rangle & \left( \langle J_{iu} J_{ui} \rangle + \frac{\mu^2}{n^2} \right) \langle [\mathsf{G}_{uu}^{(i)}]_{21} \rangle \\ \left( \langle J_{iu} J_{ui} \rangle + \frac{\mu^2}{n^2} \right) \langle [\mathsf{G}_{uu}^{(i)}]_{12} \rangle & \left( \langle J_{ui}^2 \rangle + \frac{\mu^2}{n^2} \right) \langle [\mathsf{G}_{uu}^{(i)}]_{11} \rangle \end{pmatrix} \\
+ \frac{(n-1)(n-2)}{n^2} \mu^2 & \begin{pmatrix} \langle [\mathsf{G}_{uv}^{(i)}]_{22} \rangle & \langle [\mathsf{G}_{uv}^{(i)}]_{21} \rangle \\ \langle [\mathsf{G}_{uv}^{(i)}]_{12} \rangle & \langle [\mathsf{G}_{uv}^{(i)}]_{11} \rangle \end{pmatrix},
\end{aligned}
\tag{80}
$$

with $u, v \in [n] \setminus \{i\}$ and $u \neq v$. Using that $\sigma^2 = \langle J_{ik}^2 \rangle$ and $\tau\sigma^2 = \langle J_{ik} J_{ki} \rangle$, Eqs. (80) and (80) yield Eq. (62), and consequently, in the limit of large $n$ the quantity $\mathsf{G}_{ii}$ that determines the resolvent of $\mathbf{A}$ is, neglecting subleading order terms in $n$, independent of $\mu$.

# F    Derivation of the Eq. (36) for the eigenvalue outlier $\lambda_{\mathrm{isol}}$

We derive Eq. (36) for the eigenvalue outlier of random matrices $\mathbf{A}$ as defined in Sec. 2. To this purpose, we use the cavity method for eigenvalue outliers of random matrices, as developed in Refs. [17, 27, 62], and used in Ref. [63] for the case of symmetric block matrices. Note that this method is distinct from the cavity method for the spectral density developed in [36].

Assuming $\mathbf{A}$ is diagonalisable, the matrix $\mathbf{A}$ has $n$ left and right eigenvectors denoted, respectively, by $\vec{l}_j$ and $\vec{r}_j$. Normalising left and right eigenvectors such that

$$\vec{l}_j^\dagger \vec{r}_k = \delta_{j,k}, \quad \forall j, k \in [n], \tag{81}$$

we can decompose the matrix as

$$\mathbf{A} = \sum_{j=1}^{n} \lambda_j \vec{r}_j \vec{l}_j^\dagger, \tag{82}$$

and analogously for the resolvent

$$\mathbf{G}(z) = \sum_{j=1}^{n} \frac{1}{z - \lambda_j} \vec{r}_j \vec{l}_j^\dagger, \quad \text{for} \quad z \notin \{\lambda_1, \lambda_2, \ldots, \lambda_n\}. \tag{83}$$

Consequently, for $z = \lambda_j + \eta$ it holds that

$$\lim_{\eta \to 0} \eta \mathbf{G}(\lambda_j + \eta) = \vec{r}_j \vec{l}_j^\dagger + O(\eta), \tag{84}$$

as long as $\eta$ is much smaller than the distance between $\lambda_j$ and any other eigenvalue of $\mathbf{A}$. Using the Schur complement formula Eq. (58), the Appendix F of Ref. [17] shows that the eigenvector entries $[\vec{r}_j]_k$ of $\vec{r}_j$ obey the recursion relation

$$[\vec{r}_j]_k = G_{kk}(\lambda_j + \eta) \sum_{\ell=1;(\ell \neq k)}^{n} A_{k\ell} [\vec{r}_j^{(k)}]_\ell, \tag{85}$$

in the asymptotic limit $n \gg 1$, where $\vec{r}_j^{(k)}$ is the right eigenvector of the matrix $\mathbf{A}^{(k)}$, obtained from $\mathbf{A}$ by deleting the $k$-th rows and columns, and associated with the same eigenvalue $\lambda_j$. The derivation of Eq. (85) relies on two assumptions, mainly that the eigenvalue $\lambda_j$ is well separated from other eigenvalues in the limit of large $n$, so that Eq. (84) applies, and that $\lambda_j$ is both an eigenvalue of $\mathbf{A}$ and the cavity matrices $\mathbf{A}^{(k)}$. Both assumptions are valid for eigenvalue outliers $\lambda_{\mathrm{isol}}$ in the limit of large $n$.

Setting $\lambda_j = \lambda_{\mathrm{isol}}$, we find that the entries of the resolvent in the limit of large $n$ are given by

$$G_{jj}(\lambda_{\mathrm{isol}} + \eta) = g_{21}(\lambda_{\mathrm{isol}}), \tag{86}$$

where $g_{21}$ solves Eq. (21) for $g_{22} = g_{11} = 0$ as the outlier is in the region of the complex plane outside the support set $\mathcal{S}$, and hence the trivial solution applies. Additionally, according to the law of large numbers

$$\lim_{n \to \infty} \sum_{\ell=1;(\ell \neq j)}^{n} A_{k\ell} [\vec{r}_j^{(k)}]_\ell = \mu \langle R_{\mathrm{isol}} \rangle, \tag{87}$$

where $\langle R_{\mathrm{isol}} \rangle \neq 0$ is the average value of the entries of the right eigenvector associated with the eigenvalue outlier. Substitution of Eqs. (86) and (87) in Eq. (85) yields

$$[\vec{r}_j]_k = \langle R_{\mathrm{isol}} \rangle (1 + o_n(1)), \tag{88}$$

where $o_n(1)$ denotes subleading order contributions that vanish when $n$ is large enough, and where $\langle R_{\mathrm{isol}} \rangle$ solves

$$\langle R_{\mathrm{isol}} \rangle = \mu g_{21}(\lambda_{\mathrm{isol}}) \langle R_{\mathrm{isol}} \rangle. \tag{89}$$

As for eigenvalue outliers $\langle R_{\mathrm{isol}} \rangle \neq 0$, Eq. (89) implies that the outlier eigenvalue $\lambda_{\mathrm{isol}}$ solves the Eq. (36).

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
