# Peer review of "Antagonistic interactions can stabilise fixed points in heterogeneous linear dynamical systems"

_SciPost Physics, doi:SciPost Phys. 14, 093 (2023)_

## Round 1 · Referee Report · Anonymous (Referee 1) · 2022-2-21

Strengths

1- The results presented in Eqs. (25)-(26) are a nice generalisation of the circular law for quite generic, non-symmetric random matrices.

2- Very useful recap of the different cases in Sec. 5.

3- Very well written.

Weaknesses

1- Technical aspects are sometimes taken for granted.

2 - The results presented in the manuscript are interesting but are restricted to purely linear systems.

3- This work will be of interest to experts in complex systems and theoretical ecology provided that an effort is made to better explain biological/ecological motivations.

Report

The paper is sound and well written.
The main question addressed in the manuscript concerns the possibility of stabilising a linear dynamical system through suitable interactions even when a finite fraction of variables is not stable alone. I found the connection between stability and community interaction to be quite elegant.
However, I have two main concerns about the manuscript in the current form. First, the novelty of the results needs to be better distinguished from similar results that have been obtained in close-related contexts using dynamical stability analysis and replica computations. The paper would also benefit from some discussion of how this theoretical approach might apply to real systems and experimental works.
For the sake of clarity, I would then ask the authors to get back to a few listed points and better detail them.

Requested changes

1) After Eq. (1) the authors should contextualize better and possibly add specific references while stating "Differential equations of the form (1) appear in linear stability analyses of complex systems described by nonlinear differential equations of the form [...]".

2) On page 3, the authors claim the importance of "focusing on the case of negative $\tau$ that is of particular interest for ecology" but without explaining the actual reason. I agree with this point as it allows to describe prey-predator interactions, however, I think the authors should clarify better and give more ecological motivations.

3) Concerning Sec. 2, "As will become clear later, in the limit of n ≫ 1 the leading eigenvalue is a deterministic variable that only depends on the moments of the distribution given in Eq. (3)". For an experienced researcher in disordered systems, it is immediate to get the underlying reasoning. However, I do not think it is enough for a non previously aware reader to understand why the random average can be neglected and, hence, the leading eigenvalue becomes a purely deterministic quantity (as related to the properties of the Stieltjes transform specifically at $n \rightarrow \infty$).

4) As for Eq. (20), i.e. the condition to find the boundary of the support, I would urge the authors to be more cautious as this condition has been already found in the literature using RMT techniques. Albeit in the symmetrical case, this stability bound has been precisely derived to point out marginal stability in large well-mixed ecosystems. See Biroli et al., New Journal of Physics (2018) and, more in detail, Altieri et al., SciPost Physics (2022) for a specific application of the Schur complement formula (Eq. (48) in the main text along with the derivation given in the Appendix). Same comment when introducing Eq. (21).

5) In the Conclusions, the authors claim that "A is the Jacobian matrix, which is in general different from the interaction matrix. Nevertheless, studies in, among others, ecology [10,11] and neuroscience [8,9], show that nonlinear systems do exhibit regimes with one unique stationary fixed point and random matrix theory can provide insights on system stability in this regime." I agree with the statement that, although the interaction matrix and the Jacobian (or "community matrix") are not the same matrix, their stability properties - as well as the feasibility of equilibria - are closely related, but I would remain cautious about possible generalizations. This outcome has been stressed recently in a bunch of works (see for instance, Grilli et al. (2017), Stone (2018)) according to which May's argument still applies thanks to the relationship between the Jacobian and the interaction matrix. This reasoning is valid for linear systems or generalized Lotka-Volterra models, as presented in [10], but it is not necessarily true for more complex models, for instance with sub-linear growth. With sub-linear production the stability of the community matrix is no longer directly related to that of the interaction matrix and the elements of the Jacobian matrix would strongly depend on equilibrium abundances.

Minor changes/Typos: - Before Sec. 4, "non-Hermitian" is not correctly written. - Add a punctuation mark after $z \in \mathcal{S}$ (page 7). - Just before Eq. (28), separate the two diagonal elements with a comma or '='. - In the Caption of Fig. 2, the authors mention Eqs. (28) and (29), which have not been introduced yet. Please, either move the figure downwards or introduce the two formulas to be used. - In Sec. (4.2), the authors state that "the τ = 0 case is mathematically simpler to solve". Just as a hint to the authors, this can also be seen dynamically starting from the definitions in Eqs. (1)-(3) and using the formalism from Dynamical Mean-Field Theory. The case with zero correlation implies that the contribution of the integrated response vanishes. The dynamics is thus reduced to a self-consistent process for the first two moments only of the variable $x(t)$.

  • validity: high
  • significance: good
  • originality: good
  • clarity: high
  • formatting: excellent
  • grammar: excellent

Author:  Samuel Cure  on 2022-11-14  [id 3015]

(in reply to Report 1 on 2022-02-21)
Category:
answer to question

Please find in attachment a reply to your report.

Attachment:

ref1.pdf

Anonymous on 2022-11-25  [id 3072]

(in reply to Samuel Cure on 2022-11-14 [id 3015])

I thank the authors for carefully considering all my comments and editing the draft accordingly. The implemented changes in the last version fully satisfy my previous concerns.
I am in favour of the publication of the paper in its current form.

---

## Round 1 · Referee Report · Anonymous (Referee 2) · 2022-3-16

Strengths

The authors study the stability of a large random linear system.

The underlying random matrix model considered is interesting: it is basically the sum of an elliptic model ("symmetric" entries have a prescribed tau correlation) and a random diagonal matrix.

The impact of tau in the stability of the system yields original and non trivial conclusions.

Report

I am willing to support the publication of this paper once the comments of the report (attached) are addressed.

Attachment

  • validity: high
  • significance: high
  • originality: top
  • clarity: good
  • formatting: excellent
  • grammar: good

Author:  Samuel Cure  on 2022-11-14  [id 3014]

(in reply to Report 2 on 2022-03-16)
Category:
answer to question

Please find a reply to your report in the attachment.

Attachment:

ref2.pdf

---

## Round 3 · Referee Report · Anonymous (Referee 2) · 2023-1-6

Report

I am happy with the corrections made by the authors and fully support the publication of the paper.

Requested changes

Remark: beware that references [22] and [23] are the same.

---

## Round 3 · Author Response

Dear Editor,

We apologise for the delayed response.

We took the time to carefully address all points raised by the Referees.

In addition, we have included exact analytical results on the leading eigenvalue when the the mean value of the matrix entries is nonzero. This result was missing in the previous manuscript as it involves a different mathematical technique involving outliers, and hence we put, admittedly somewhat artificial, the mean value of the matrix entries to zero. However, in the new version of the manuscript we provide exact analytical results for the eigenvalue outliers when the mean values of the matrix entries are nonzero, and we also discuss the effect of antagonistic interactions on linear stability when the leading eigenvalue is an outlier.

Given these improvements, we hope the paper will be considered favourably for publication in SciPost Physics.

In the reply to the referee, you can find a detailed reply to all the Referee's comments [Referee comments in blue and our reply in black].

Kind regards,
The Authors

---

## Round 3 · List of Changes

• Added Section 5;
  • Added Appendices E and F;
  • Added Figures 5 and 7;
  • Clarified derivation steps in Appendix A;
  • Included several references;
  • Changed the title and abstract following comments of Referee B;
  • We have taken the comments of the Referees onboard and made several local changes in the manuscript (see response to referee reports).

---

## Editorial Decision

published